# Effects of Three Cultural Practices on *Drosophila suzukii* (Diptera: Drosophilidae) in Open Blueberry Fields in Florida

**DOI:** 10.3390/insects13100957

**Published:** 2022-10-20

**Authors:** Albertha J. Parkins, Muhammad Haseeb, Oscar E. Liburd, Lambert H. B. Kanga

**Affiliations:** 1College of Agriculture and Food Sciences, Center for Biological Control, Florida Agriculture and Mechanical University, Tallahassee, FL 32307, USA; 2Department of Entomology, University of Georgia, Tifton, GA 31794, USA; 3Entomology and Nematology Department, University of Florida, Gainesville, FL 32611, USA

**Keywords:** invasive pest, damage, population, cultural practices, pest management

## Abstract

**Simple Summary:**

The spotted-wing Drosophila is an invasive pest of berry fruits in Florida. The pest is difficult to control with pesticides, and resistance is developing. The current study was carried out in north Florida to see if non-pesticidal pest control tactics using three mulching practices could reduce its population density to increase growers’ berries productivity. The experiments were conducted for two years while using two plant-based mulches (short pine needles and pine bark) and one fabric-based mulch in the open blueberry fields. In general, the fly population was reduced within the selected pine bark and black weed fabric mulches. This is the first report of such practices deployed to manage the spotted-wing Drosophila in Florida. We are confident that these selective cultural practices could be utilized in open blueberry fields before berries settings to reduce pest pressure for both conventional and organic blueberry growers in Florida. Indeed, this will increase growers’ berry production and profitability.

**Abstract:**

The spotted-wing Drosophila (SWD), *Drosophila suzukii*, is native species in Southeast Asia. For over a decade, this invasive pest has been globally expanding. The economic losses to soft fruits and stoned fruits in the United States are increasing every year. Presently, the only viable tool to reduce the SWD population is the continued use of broad-spectrum insecticides. Pesticide resistance is appearing in the populations for the SWD. Organic farmers have limited options to control this pest in open fields. The major goal of this study was to develop cost-effective pest management strategies to manage the SWD using three types of mulches (two plant-based and one fabric-based) to reduce fly population and damage in open blueberry fields in north Florida. The study was conducted in two fruiting seasons (2017 and 2018). The study results demonstrated that the fly trap catches in 2017 shortleaf pine needle mulch had much higher populations (about 2.5-fold) of the SWD than all other treatments. In 2018, the numbers were about 1.7-fold more on shortleaf pine needle mulch than on other treatments. The fine texture of the mulch (pine needles) can easily facilitate the emergence of the SWD if the mulch is not thick enough. Although the pine needles covered the soil surface, it may have been too thin and thus allowed the SWD adults to emerge from the soil without much hindrance. In 2018, a higher population of the SWD was recorded from all the mulching practices. However, there were no significant differences in trap catches between all treatments. In general, the fly population is reduced with the use of pine bark and black weed fabric mulches. This is the first study that reports the effects of three mulches in controlling the SWD populations, which could benefit conventional and organic blueberry growers.

## 1. Introduction

*Drosophila suzukii* (Matsumura) (Diptera: Drosophilidae), commonly known as the spotted-wing Drosophila (SWD), is an exotic, polyphagous insect pest native to Asian origin [1,2]. It is a small fly that is currently impacting the production of major berry crops such as blueberries, strawberries, caneberries (raspberries and blackberries), and cherries in the United States. In Florida, the SWD is a very serious pest of blueberries. This pest differs from common vinegar flies within the genus *Drosophila* because the female actively seeks and successfully attacks healthy (undamaged) fruits by laying her eggs inside the fruit [3,4]. The adults have a pale brown to yellowish-brown thorax with black bands on the abdomen. The antennae are short and stubby with branched arista. Males have a distinguishing dark spot along the front edge of each wing. Female has a distinct serrated ovipositor which helps the fly to cut ripened berry fruits while laying eggs (Figure 1A,B). The SWD is a significant concern because of its ability to persist in the ripening fruits, high reproductive capacity, short life cycle, the ability persists in diverse climates and weather conditions, polyphagy, and limited effective natural enemies [3]. Consequently, *D. suzukii* populations can damage several different varieties of ripening fruit during a single growing season under optimum environmental conditions. Additionally, it may attack several non-berry fruits (stone fruits) including cherries, peaches, and plums. The life cycle of the SWD is completed in 7–14 days, depending on climatic conditions [5].

For almost a decade, researchers and growers have worked to develop integrated pest management (IPM) strategies to limit damage and losses associated with the SWD [6]. A key component of a successful IPM program is monitoring and early detection of this pest, leading to improved decision-making, more efficient timing, and quantitative evaluation of treatments.

The control of the SWD is accomplished mainly by chemical management in the United States. Due to extensive use of synthetic insecticides in the blueberry crops the risk of pesticide resistance, and the negative effects on non-target organisms, such as pollinators and natural enemies are increasing. Therefore, the development of alternative management strategies is essential [7] to support growers and production systems.

Cultural control is one of the most usable methods of IPM for natural and organic growers. Additionally, the demand for natural products to manage pests is increasing in recent years [8,9]. Cultural control involves horticultural management practices such as cultivar selection, mulching, early harvest, sanitation, burying, pruning, and mass trapping of the SWD [10,11,12]. Cultural control may involve selecting a cultivar that completes harvest before fly populations begin to build up [13], and early-ripening varieties of blueberries such as duke and draper can help decrease the chances of heavy fly infestation [14,15].

Mulching is a horticultural tactic that could help control the SWD and has a beneficial effect on blueberry cultivation [16]. Mulching provides horticultural benefits such as weed suppression and enhanced plant growth by reducing moisture evaporation from the soil, increasing water filtration, improving soil structure, enhancing the availability of mineral nutrients, and helping to acidify the soil and moderate the soil temperature [17,18,19,20]. Previous studies showed that mulching practices have successfully captured other dipteran flies, such as Dolichopodidae and Mycetophilidae. It is also effective to control other insects, such as whiteflies. Reflective mulches have been shown to be more effective in repelling whiteflies from plants compared to white and black polyethylene mulches [21]. A field experiment carried out at the University of Florida on the effect of organic mulches on soil surface insects, and other arthropods concluded that Diptera was more affected (*p* ≤ 0.05) by pine bark mulch than by mulches of sunn hemp and sorghum Sudan grass [22].

## 2. Materials and Methods

### 2.1. Blueberry Study Site

The study was conducted in an open blueberry field at the North Florida Research and Education Center (UF/IFAS), Quincy, Florida [30°32′25″ N, 84°35′23″ W]. The Rabbiteye blueberry field was established in 2003. At this site, thirty plants of Austin, Brightwell, Climax, Powderblue, and Premier Cultivars have been planted at 1.5 m within and 3.05 m between rows, respectively. The study was carried out during the summer months (May–July) of 2017 and 2018.

### 2.2. Experimental Design

During the two years study period, four experimental units were randomly arranged within a single block. A total of two blocks were assigned with replications made annually. The treatments included weed fabric mulch, shortleaf pine needle mulch, pine bark mulch, and untreated control. The four treatments were randomly arranged within each block. Three mature blueberry buffer plants were used to separate each treatment (the average planting distance between the plants was 1.5 m). Adult SWD activity was monitored using two attractant lures used alongside Scentry^®^ traps (Scentry Biological Inc., Billings, MT, USA). Trap 1 contained a Scentry bait pouch lure (herein referred to as bait trap) while traps 2 contain a liquid bait lure (herein referred to as Suzukii bait trap). The traps were distributed over an area of 137 m^2^ at each sample site. Adult SWD caught in a trap in each treatment were recorded over a period of twelve weeks. The experiment took place in the spring months (March–May) in 2017 and 2018.

### 2.3. Types and Application of Mulches

Types of mulches used in this study were: weed fabric mulch, pine bark mulch, shortleaf pine needle mulch, and a control (no mulch). The weed fabric mulch was placed around the base of each blueberry plant. A fabric cutter was used to a small line in the fabric close to the plant allowing it to fit around the base of the plant, after which metal pins were hammered into the soil to pin the fabric the soil. After the weed fabric was placed around the six plants, two plants were left as buffer plants to create space showing the transition from weed fabric to pine bark mulch.

The pine bark mulch was placed around the base of each blueberry plant. Approximately one bag (2 cu. ft.) was used for every two plants, three bags per row, for a total of six bags. Then, two plants were skipped, which served as buffer plants showing a transition from pine bark mulch to shortleaf pine needle mulch. 

The shortleaf pine needle mulch was placed at the crown of each blueberry plant. Approximately 1.50 bales of pine needles were used for three plants. Hence, three bales were used for the six plants per treatment. Then, two plants were skipped, which served as buffers showing a transition from shortleaf pine needle mulch to the control. No mulch was used in control (untreated). Six plants in each treatment were used for the control.

### 2.4. Monitoring Traps

A total of eight traps were installed in the field after mulching was completed: four were Scentry^®^ Traps (Scentry Biological Inc.), a user-friendly monitoring system with a pouch lure that is used for early detection of the SWD (Greatlakes IPM, Vestaburg, MI, USA). This trap is mostly red externally with the lure hanging inside, small holes at the sides, and about one inch of water added as a drowning solution. The remaining four traps contained Suzukii bait, a food attractant designed to capture the SWD. The Suzukii bait was placed in modified transparent plastic trap containers (each 900 mL with 27 holes (each 4 mm) with three entry point for fly entrance). Each container was modified by placing a red color tape (3 M Polyethylene Film Tape) around to attract the SWD (Figure 2). An earlier study reported that the SWD was attracted to dark colors [23]. The traps were placed in the canopy areas of each treated blueberry plant. It is recommended that Scentry^®^ bait lures be changed every four to six weeks. In contrast, the liquid bait lure can last through the entire season. Regardless, all traps were serviced (bait lure, liquid lure, and trap cleaned) every six weeks to avoid bias. In addition, bait trap drowning/killing solution was replaced weekly.

The first collection of the SWD flies was carried out eight days after the traps were set up. After this, sampling was carried out once a week. Sampling continued for 12 weeks until all blueberry fruits were harvested. Flies were collected from traps by straining them through the plastic mesh and placed in vials containing 95% ethanol for preservation. Flies were taken to the laboratory and identified under a microscope using morphological characteristics of the species. The total number of male and female flies captured in each treatment was recorded.

### 2.5. Statistical Analyses

Data on the efficacy of the types of mulch for the control of spotted-wing drosophila adults captured in the traps were analyzed using repeated measures analysis (PROC Mixed) [24]. Treatments were modeled as fixed effects, and gender-by-treatment interactions were modeled as random effects. The studies were conducted in single experimental units; therefore, time (weeks/year) was used to provide the replications to test the fixed effects of the treatments. Similarly, time (year 1 and year 2) provided the replications to test the fixed effects of the efficacy of the type of traps (Proc *t*-test) [24]. The means of the treatments were separated by Tukey’s Studentized range test [24]. Data were transformed to log (*x* + 1) to satisfy the assumptions of normality before analysis, and normality was confirmed after data transformation. A significance level of alpha = 0.05 was used for all statistical tests.

## 3. Results

### 3.1. Suzukii^®^ Bait Trap

Over the 12-week experimental periods in 2017, there were significant differences among the types of mulch tested (*F* = 3.11; *df* = 3, 44; *p* = 0.0359). However, the gender by-treatment interactions were not significant (*F* = 0.29; *df* = 3, 44; *p* = 0.8318) (Table 1). The highest numbers of adult SWD were captured from the shortleaf pine needle mulch treatments. The pine bark mulch and weed fabric mulch captured similar numbers of the SWD as did the control (Table 1). The type of mulch did not have any significant impact on the number of male or female adult SWD captured in the traps (*F* = 1.04; *df* = 1, 22; *p* = 0.3198) (Table 2).

There were no significant differences among the different mulch types tested in 2018 (*F* = 0.92; *df* = 3, 44; *p* = 0.4382) (Table 1). Additionally, the gender by-treatment interactions were not significant (*F* = 0.55; *df* = 3, 44; *p* = 0.6499). Over the 12-week period of the experiments, there were significantly higher numbers of females than male flies caught in the Suzukii bait trap (*F* = 4.88; *df* = 1, 22; *p* = 0.0379) (Table 2).

The seasonal patterns of occurrence of the SWD varied over time depending on mulch practices and the gender of the flies. During the twelve-week sampling period in 2017, the total trap catches in the shortleaf pine needle mulch were 735 flies, in weed fabric mulch 293 flies, in control 289 flies, and in pine bark mulch 284 flies (Figure 3). The relative abundance of males and females in the trap catches was similar. The numbers of flies were low during the first three weeks of sampling but increased from week four to eight in all treatments. However, the number of flies declined during the 9th and 10th week of sampling. During the last two weeks (when blueberry fields were being harvested), the number of flies decreased and were absent by week 12 (Figure 3).

During the following year in 2018, the total trap catches over the twelve weeks of sampling were 981 flies in the shortleaf pine needle mulch, 583 flies in control, 495 flies in the pine bark mulch, and 312 flies in the weed fabric mulch (Figure 4). There were significantly more females than males captured (Table 2), which suggested that females may be more attractive to the lure composition in the Suzukii^®^ bait trap. Low numbers of flies were collected during the first four weeks of sampling. The highest numbers of flies were collected during the sixth to eighth weeks in all treatments. As in 2017, the number of flies declined during the 9th and 10th week of the sampling periods. During the last two weeks, the number of flies decreased with the harvesting of the blueberries (Figure 3). The seasonal dynamics of the populations were similar to those in 2017. The numbers of flies caught in the Suzuki bait traps in 2017, and 2018 were not significantly different during the twelve-week sampling periods (*t* = −0.13; *df* = 22; *p* = 0.8948). However, the numbers of *D. suzukii* collected in 2018 were relatively higher than in 2017 (Table 3). This showed that although the fly numbers have been increasing gradually in two years, the density was not significantly different between the two years.

### 3.2. Scentry^®^ Trap

The type of mulch did not have any significant effect (*F* = 1.19; *df* = 3, 44; *p* = 0.3239) on the numbers of flies captured in the Scentry^®^ traps in 2017 (Table 4); the treatment by-gender interactions were also not significant (*F* = 0.77; *df* = 3, 44; *p* = 0.5171). Although, the number of flies in pine bark mulch and feed weed fabric mulch in both years were relatively low compared with the other treatments. The type of mulch used did not have any significant effect on the numbers of male and female adults of *D. suzukii* captured in the traps (*F* = 0.10; *df* = 1, 22; *p* = 0.7519) (Table 5).

In 2018, there were no significant differences among the mulch types tested (*F* = 0.03; *df* = 3, 44; *p* = 0.9936) (Table 4). The gender by-treatment interactions were not significant (*F* = 0.26; *df* = 3, 44; *p* = 0.8567). We also found that the mulch type did not have any significant effect on the numbers of male and female adults of *D. suzukii* captured in the traps (*F* = 0.05; *df* = 1, 22; *p* = 0.8243) (Table 5).

The numbers of adult *D. suzukii* captured during the twelve-week sampling period in 2018 were significantly higher than those in 2017 (*t* = −2.59; *df* = 22; *p* = 0.0168) (Table 6).

In 2017, the Scentry^®^ trap captured 821 flies in the shortleaf pine needle mulch, 579 flies in control, 409 flies in the weed fabric mulch, and 159 flies in the pine bark mulch during the twelve-week sampling period (Figure 4). For the season, we found low numbers of flies during the first four weeks of collection. The highest numbers of flies were captured between the 6th and 10th week. During the last two weeks, the number of flies declined (Figure 5).

In 2018, the Scentry^®^ trap captured 2956 flies in the pine bark mulch, 2386 flies in the weed fabric mulch, 2121 flies in the control and 1856 flies in the shortleaf pine needle mulch (Figure 4). Seasonally, the fly populations were low during the first three weeks of sampling and increased during the fourth and fifth week. The highest numbers of flies occurred during the period from six to ten weeks. Fly populations declined during the last two weeks of sampling (Figure 6).

### 3.3. Trap Comparison

The total numbers of adult *D. suzukii* captured over two years in Suzukii^®^ bait traps did not differ significantly from those caught in Scentry^®^ traps (*t* = 0.87; *df* = 2; *p* = 0.4742). However, Scentry^®^ traps collected relatively more flies than Suzukii traps during the experimental periods (Table 7).

## 4. Discussion

The results of this study demonstrated that Suzukii bait trap in 2017 shortleaf pine needle mulch had much higher populations (about 2.5-fold) of the SWD than all other treatments. In 2018, the numbers were about 1.7-fold more on shortleaf pine needle mulch than on other treatments. The fine texture of the mulch (pine needles) can easily facilitate the emergence of the SWD if the mulch is not thick enough. Although the pine needles covered the soil surface, they may have been too thin and thus allowed the SWD adults to emerge from the soil without much hindrance. In 2018, a higher population of the SWD was recorded from all the mulches tested; however, there were no significant differences in trap catches between all treatments. The data for Suzukii bait traps in 2017 and 2018 indicated that pine bark mulch and weed fabric mulch had the lowest trap catches. Pine bark is similar to wood chips covering some of the soil surfaces and should serve as a barrier, reducing the emergence of the SWD. Similarly, weed fabric covers the soil surface where applied and should prevent the emergence of the SWD from those areas. Weed fabric mulch can reduce the SWD populations in two ways: heat radiating from the weed fabric during the day can kill larvae falling out of berries on the fabric mulch, and weed fabric also forms a barrier preventing flies from pupating in the soil or preventing emergence from the soil

In 2017, Scentry^®^ traps placed over pine bark mulch captured the lowest number of *D. suzukii*. Pine bark mulch had 5-fold fewer SWD than the shortleaf pine needle mulch. As was shown for Suzukii traps, it appears that pine bark mulch can be used to provide some degree of reduction in the populations of the SWD. The constituents of pine bark and its ability to cover some areas of the soil which prevent the SWD from pupating may contribute to the suppression of the SWD populations. Pine bark has shown to become hydrophobic when allowed to dry to a moisture content below 34% by volume [25,26]. A reduced mulch humidity leads to unfavorable conditions for the SWD growth and development and may have contributed to the low populations and emergence rates. However, additional studies are needed to verify this finding and determine whether or not the pine bark mulch can be recommended as a successful control measure for the SWD in open field conditions. Additionally, short buffer zones (6 m) were kept in between the treatments under this study, and an earlier study could be limiting factor to provide clear picture of the actual adult flies caught as affected by each treatment [27]. In another study a buffer zone of 15 m was kept between the treatments to capture adult flies of the *D. suzukii* [28].

The data collected in 2017 and 2018 for all traps showed more females than males. These results are supported by an earlier study [29] which reported that Scentry^®^ traps release fermented wine volatiles that is more attractive to females than males. We did not find any significant differences in terms of trap catches between Suzukii bait traps and Scentry^®^ traps, although the highest total number of flies was caught in Scentry^®^ traps. Traps are generally used to monitor population trends and determine when SWD flies are present in the fields.

The total number of adult flies captured throughout the experimental periods indicated that the population of the SWD in 2018 was higher than in 2017. This difference in population numbers between years could be due to the lack of pruning of blueberry bushes in 2018. More fruits on blueberry bushes would provide more resources to support higher fly populations. In addition, more bushes provide shade that would protect the SWD from direct rays of the sun and natural enemies. Further, more canopy cover can increase humidity and provide conditions that are conducive to the SWD growth and development [15]. We suggest that the increase in the density of fly populations was due to the availability of food, environmental conditions, and residual flies from the previous year. However, several other factors may contribute to the differences in the populations, such as the softness of the skin of the blueberries, varieties ripening at different time intervals, and insufficient mulch coverage of areas beneath the bushes. Dipteran flies such as the SWD are extremely good fliers and have a high dispersal ability. Therefore, the pest can colonize and persist for a longer period of time under a wide range of environmental conditions, including extremes of temperature and humidity. A laboratory study reported that the SWD flies were found to avoid highly saturated environments where relative humidity (RH) exceeded 87% [12,29]. Additionally, recently, mulching practices were utilized in blueberries and other fruits against the adult fly emergence with some success [30,31]. Further study is critical to see if these practices could be deployed in the bigger blueberry fields in Florida and other ecologies.

## 5. Conclusions

The use of three mulching practices in open blueberry fields provided an insight into limiting the population density of *Drosophila suzukii*. This is the first study in this direction in Florida that showed that cultural practices could reduce the density of the spotted-wing Drosophila in blueberries. Two mulches in particular, including pine bark and weed mulch, showed potential as the fly population was reduced in general. Indeed, this information could be applicable to integrated pest management in situations where entomologists could detect, monitor and alter habitats to reduce pest pressure and increase the crop productivity of conventional and organic blueberry growers in Florida. However, this study provides preliminary information on the effects of three selected mulches on the capture of *Drosophila suzukii* (a highly mobile insect) due to the very short buffer zone between the treatments. Therefore, additional study in the bigger and open blueberry fields is necessary to elaborate on these results.

## Figures and Tables

**Figure 1 insects-13-00957-f001:**
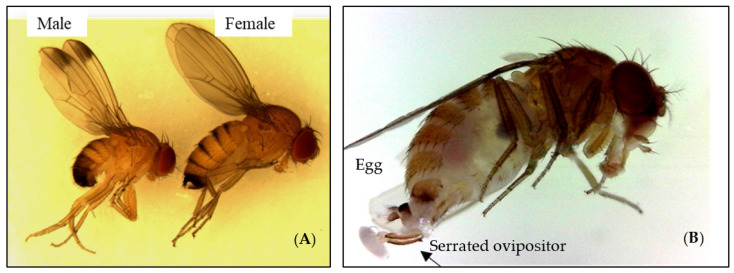
(**A**) male (wings with spots at the apex) and female (no wing spots) adults of *D. suzukii*, (**B**) Arrow showing the serrated ovipositor with rice-shaped egg.

**Figure 2 insects-13-00957-f002:**
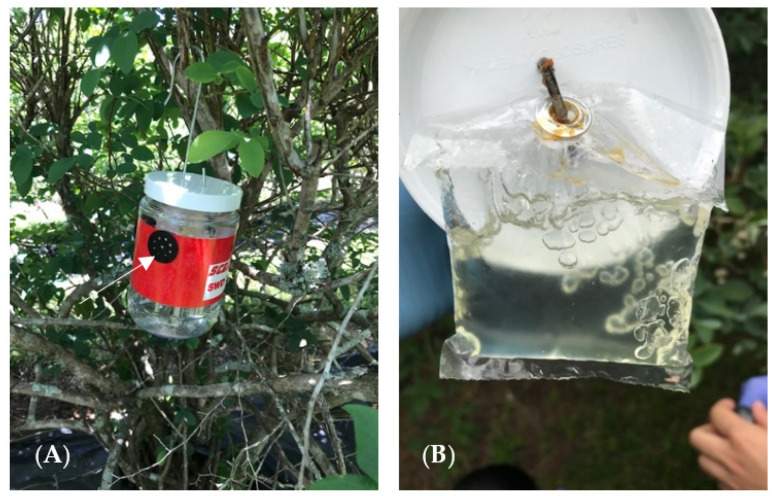
(**A**) Image of a Scentry^®^ trap with red color tape, and arrow showing holes for adult fly entrance; (**B**) Lure, used in this study.

**Figure 3 insects-13-00957-f003:**
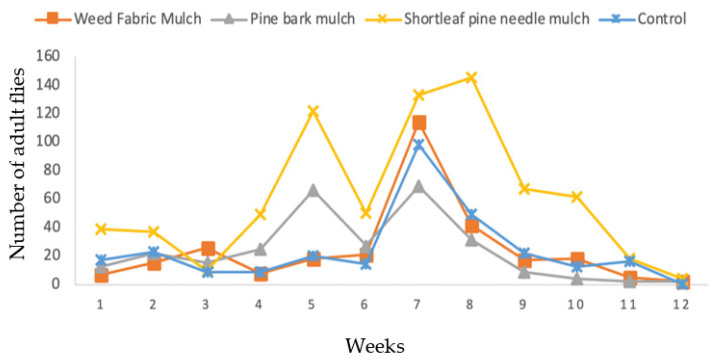
Effects of three mulching practices in comparison to unmulched control on the capture of *D. suzukii* in the blueberry field using Suzukii^®^ bait trap in 2017.

**Figure 4 insects-13-00957-f004:**
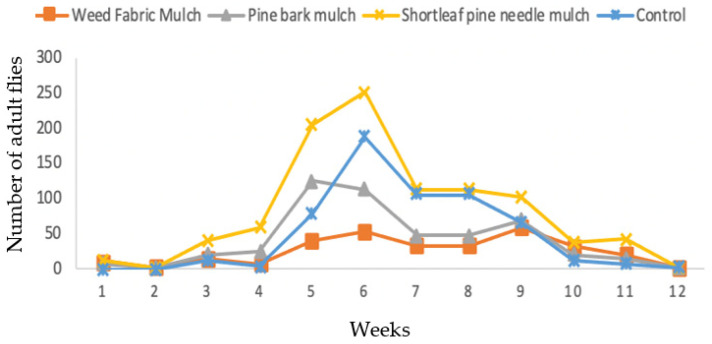
Effects of three mulching practices in comparison to unmulched control on the capture of *D. suzukii* in the blueberry field using Suzukii^®^ bait trap in 2018.

**Figure 5 insects-13-00957-f005:**
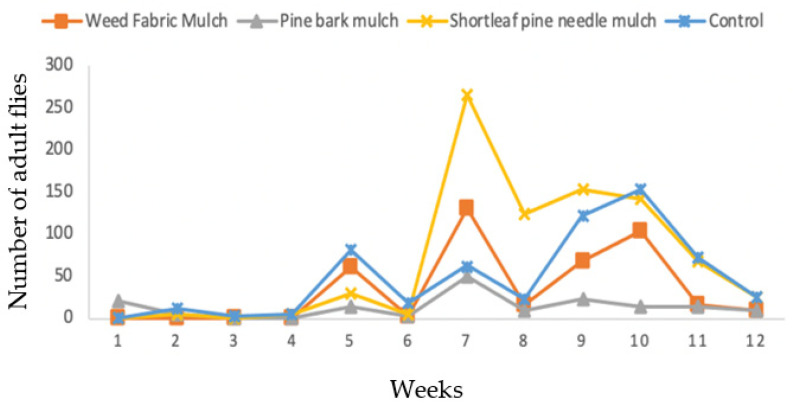
Effects of three mulching practices in comparison to unmulched control on the capture of *D. suzukii* in blueberry field using Scentry^®^ trap in 2017.

**Figure 6 insects-13-00957-f006:**
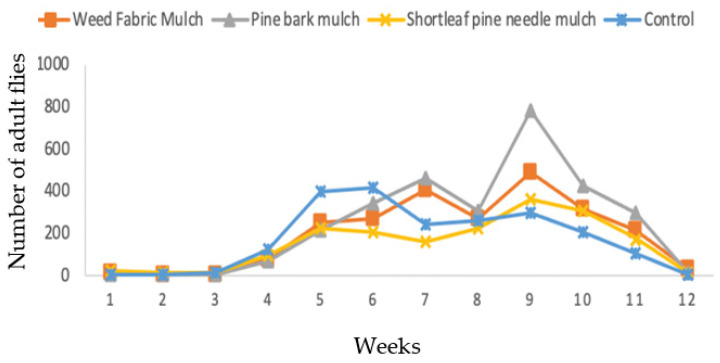
Effects of three mulching practices in comparison to unmulched control on the capture of *D. suzukii* in the blueberry field using Scentry^®^ trap in 2018.

**Table 1 insects-13-00957-t001:** Average numbers of adult *D. suzukii* caught on blueberries in different mulching practices using Suzukii^®^ bait traps during the sampling periods in 2017 and 2018.

2017			2018		
Treatments	N ^1^	Mean ± SE	Treatments	N ^1^	Mean ± SE
Shortleaf pine needle mulch	12	61.25 ±13.76 ^a^	Shortleaf pine needle mulch	12	81.75 ± 23.05 ^a^
Weed fabric mulch	12	24.42 ± 8.71 ^b^	Control	12	48.58 ± 17.49 ^a^
Control	12	24.08 ± 7.54 ^b^	Pine bark mulch	12	41.25 ± 12.08 ^a^
Pine bark mulch	12	23.67 ± 6.55 ^b^	Weed fabric mulch	12	26.00 ± 5.58 ^a^

^1^ Sampling weeks. Means in column are not significantly different if followed by the same letter (*p* > 0.05).

**Table 2 insects-13-00957-t002:** Average numbers of adult males and females caught on blueberries using Suzukii^®^ bait traps during the sampling periods in 2017 and 2018.

2017			2018		
Gender	N ^1^	Mean ± SE	Gender	N ^1^	Mean ± SE
Female	12	73.58 ± 15.99 ^a^	Female	12	148.00 ± 39.50 ^a^
Male	12	59.83 ± 18.91 ^a^	Male	12	50.33 ± 18.32 ^b^

^1^ Sampling weeks. Means in the column are not significantly different if followed by the same letter (*p* > 0.05).

**Table 3 insects-13-00957-t003:** Average numbers of adult *D. suzukii* collected on blueberries using Suzukii^®^ bait trap during the sampling periods in 2017 and 2018.

Year	N ^1^	Mean ± SE
2017	12	133.42 ± 32.98 ^a^
2018	12	197.58 ± 55.72 ^a^

^1^ Sampling weeks. Means in the column are not significantly different if followed by the same letter (*p* > 0.05).

**Table 4 insects-13-00957-t004:** Average numbers of adult *D. suzukii* caught on blueberries with different mulching practices using Scentry^®^ traps during the sampling periods in 2017 and 2018.

2017			2018		
Treatments	N ^1^	Mean ± SE	Treatments	N ^1^	Mean ± SE
Shortleaf pine needle mulch	12	68.42 ± 24.53 ^a^	Pine bark mulch	12	246.33 ± 25.38 ^a^
Control	12	48.25 ± 14.54 ^a^	Weed fabric mulch	12	198.83 ± 13.08 ^a^
Weed fabric mulch	12	34.08 ± 13.05 ^a^	Control	12	176.75 ± 19.28 ^a^
Pine bark mulch	12	13.25 ± 3.85 ^a^	Shortleaf pine needle mulch	12	154.67 ± 16.93 ^a^

^1^ Sampling weeks. Means in column are not significantly different if followed by the same letter (*p* > 0.05).

**Table 5 insects-13-00957-t005:** Average numbers of adult males and females caught on blueberries using Scentry^®^ traps during the sampling periods in 2017 and 2018.

2017			2018		
Gender	N ^1^	Mean ± SE	Gender	N ^1^	Mean ± SE
Female	12	92.75 ± 28.48 ^a^	Female	12	477.92 ± 120.66 ^a^
Male	12	71.33 ± 23.73 ^a^	Male	12	294.92 ± 66.01 ^a^

^1^ Sampling weeks. Means in column are not significantly different if followed by the same letter (*p* > 0.05).

**Table 6 insects-13-00957-t006:** Average numbers of adult *D. suzukii* collected on blueberries using Scentry^®^ traps during the sampling periods in 2017 and 2018.

Year	N ^1^	Mean ± SE
2017	12	164.00 ± 47.34 ^a^
2018	12	776.58 ± 224.17 ^b^

^1^ Sampling weeks. Means in the column are not significantly different if followed by the same letter (*p* > 0.05).

**Table 7 insects-13-00957-t007:** Comparative trap catches of adult *D. suzukii* on blueberries using Suzukii traps and Scentry^®^ traps in 2017 and 2018.

Type of Trap	N ^1^	Mean ± SE
Suzukii	2	1986.02 ± 386.13 ^a^
Scentry^®^	2	5643.50 ± 3686.48 ^a^

^1^ Sampling years. Means in column are not significantly different if followed by the same letter (*p* > 0.05).

## Data Availability

The data supporting this study’s findings are available from the corresponding author: Muhammad Haseeb, upon reasonable request.

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
