# Peer review of "Effects of Three Cultural Practices on Drosophila suzukii (Diptera: Drosophilidae) in Open Blueberry Fields in Florida"

_insects, 2022, doi:10.3390/insects13100957_

Round 1

Reviewer 1 Report

Major Comment:

The study design is confusing. The sampling size and the sampling area were not satisfactory. How the authors ensured that the sampling was not overlapped for a study involving an organism such as an actively flying Drosophila. The distance between the study group was not sufficient to ensure that there is no overlapping between the groups.

Introduction

Line 10- ability “to” persist

Line 11- delete-  “,  among  other  characteristics”

Line 12- mention a few other non-berry fruits

Line 22-25: “Due to restrictions on the use of synthetic insecticides in target crops, the risk of pesticide resistance,  and the negative effect on non-target organisms, such as pollinators and natural enemies, the development of alternative management strategies is essential [7] to support growers and production systems.”- The sentence is very complex. Split into simple sentences.

Rather than the term restrictions, it is better to use extensive

Line 26-27: “Cultural control is one of the most effective methods of IPM when it comes to natural and organic growers, especially with the increase in demand for natural products  [8,9].” – Rephrase the sentence.

Materials and Methods

2.3. Types and Application of Mulches

Line 13: “Approximately 1.50 bales of pine needles were used for three plants. Hence six bales were used for the two rows”. If 1.50 bales can be used for three plants, then a total of 3 bales should be used for 6 plants instead of 6 bales. Please clarify.

Need to detail the Suzuki trap used in the study. How it was modified?

Results

The interpretation of the results was too confusing. Make it simpler

Why are the specific traps included in the study? 

Author Response

Major Comment:

The study design is confusing. The sampling size and the sampling area were not satisfactory. How the authors ensured that the sampling was not overlapped for a study involving an organism such as an actively flying Drosophila. The distance between the study group was not sufficient to ensure that there is no overlapping between the groups.

Response: We agree with the observation that highly mobile insect like Drosophila suzukii could move between the treatments. However, we made a buffer zone of three plants between each treatment to avoid this. Also, we understand invader adult flies might also enter the berry blocks we covered in this study.

Introduction

Line 10- ability “to” persist

Response: Modified

Line 11- delete-  “,  among  other  characteristics”

Response: Deleted

Line 12- mention a few other non-berry fruits

Response: A new sentence “Also, it may attack several non-berry fruits (stone fruits) including cherries, peaches, and plums” has been inserted.

Line 22-25: “Due to restrictions on the use of synthetic insecticides in target crops, the risk of pesticide resistance, and the negative effect on non-target organisms, such as pollinators and natural enemies, the development of alternative management strategies is essential [7] to support growers and production systems.”- The sentence is very complex. Split into simple sentences.

Response: Modified to “Due to extensive use of synthetic insecticides in the blueberry crops, the risk of pesticide resistance, and the negative effects on non-target organisms, such as pollinators and natural enemies are increasing. Therefore, the development of alternative management strategies is essential [7] to support growers and production systems”

Rather than the term restrictions, it is better to use extensive

Response: Extensive inserted.

Line 26-27: “Cultural control is one of the most effective methods of IPM when it comes to natural and organic growers, especially with the increase in demand for natural products  [8,9].” – Rephrase the sentence.

Response: The sentence has been rephrased to “Cultural control is one of the most usable methods of IPM for natural and organic growers. Also, the demand for natural products to manage pests is increasing in recent years [8,9].

Materials and Methods

2.3. Types and Application of Mulches

Line 13: “Approximately 1.50 bales of pine needles were used for three plants. Hence six bales were used for the two rows”. If 1.50 bales can be used for three plants, then a total of 3 bales should be used for 6 plants instead of 6 bales. Please clarify.

Response: Thank you. The statement has been modified to 3 bales.

Need to detail the Suzuki trap used in the study. How it was modified?

Response: A detail is provided with a photo of the trap used in the study.

Results

The interpretation of the results was too confusing. Make it simpler

Response: Where applicable, we have made the results simpler for the readers.

Why are the specific traps included in the study? 

Response: In our earlier study by Harmon et al. 2019 (https://www.mdpi.com/2075-4450/10/10/313), we found that Scentry trap collected more flies than other traps in open blueberry fields. Therefore, we used Scentry type of trap to carry out this study.

Reviewer 2 Report

This is an interesting study examining the impact of different mulching practices on numbers of SWD collected in traps in blueberries. Variables included mulch types and trap/lure combinations.  This is an excellent research question to approach and certainly of interests to blueberry producers.  The study is well done and well discussed but needs clarification in some places.  Also, I think that a few more statistical comparisons at a couple of selected points would highlight results that strengthen the manuscript.

I am completely confused about what you used as traps. You state that you used 4 Scentry traps with a lure and then you state that you used a Suzukii lure. It seemed that what you are saying is that you used 8 traps (made by Scentry) and that four of those traps had a Scentry lure and four of them had a Suzukii liquid bait lure (Bioiberica). But I see later on that you say that you used modified trap containers to which you added red tape.  What type of traps (brand?), where are the openings, is it similar in size to the Scentry traps?  Where they traps by Suzukii or another brand (in which case are you referring to them as Suzukii traps when they are some other traps with Suzukii brand attractant? Are these traps that are associated with the Suzukii liquid bait and company? Are they generally the same size and shape with roughly similar openings to the Scentry ones? With similar amount of red visual stimuli?  Please rewrite this so that it is really clear what you actually did.  This needs to be clearly stated.

“trap is mostly red externally with the lure hanging inside, small holes at the sides” I think that it is important to paint a better picture of the trap for the reader.  It is a clear jar covered with a primarily bright red label.  Add the city and state for the suppliers, please.

Great Lakes IPM, should have city and state

Provide the company names, city and state for the two types of traps please

“The experiments included two recommended traps” I don’t think that you mean that the experiments included the traps but that the numbers of SWD in each treatment were monitored by two different types of traps.

Suggested wording in title for table 1. “Average numbers of adult D. suzukii caught on blueberries with different mulching practices” rather than “Average numbers of adult D. suzukii caught on different mulching practices in blueberries”

You have an inconsistent use of the ® after Suzukii trap in the manuscript.

You stated that by week 11 and 12 the blueberries may have been harvested.  When is the time of optimal attraction of SWD to blueberries?  Presumably it is when they are ripening so that earlier in the season when the berries are small, there may be fewer volatiles to attract and retain the flies in the area and that would reflect lower numbers as well.  I wonder how the statistical comparisons would look if you analyzed the data only making comparisons from weeks when blueberries were ripening or when your overall numbers (weekly collection) exceeded maybe 30 or so.  I think that the initial low collections are complicating your statistical comparisons. The high SE results from including early and late season data that have very low collections. I think that if examine some specific dates when flies are a high populations, that you would see difference in collections between the mulch types.

Do the larvae not pupate in the mulch?  It may be that they do better pupating in the pine needle mulch than the others.

Figure legends.  ‘Effects of three mulching practices on the capture of ….’ Should be ‘Effects of three mulching practices in comparison to unmulched controls on the capture of….’

Author Response

Comments and Suggestions for Authors

This is an interesting study examining the impact of different mulching practices on numbers of SWD collected in traps in blueberries. Variables included mulch types and trap/lure combinations.  This is an excellent research question to approach and certainly of interests to blueberry producers.  The study is well done and well discussed but needs clarification in some places.  Also, I think that a few more statistical comparisons at a couple of selected points would highlight results that strengthen the manuscript.

I am completely confused about what you used as traps. You state that you used 4 Scentry traps with a lure and then you state that you used a Suzukii lure. It seemed that what you are saying is that you used 8 traps (made by Scentry) and that four of those traps had a Scentry lure and four of them had a Suzukii liquid bait lure (Bioiberica). But I see later on that you say that you used modified trap containers to which you added red tape.  What type of traps (brand?), where are the openings, is it similar in size to the Scentry traps?  Where they traps by Suzukii or another brand (in which case are you referring to them as Suzukii traps when they are some other traps with Suzukii brand attractant? Are these traps that are associated with the Suzukii liquid bait and company? Are they generally the same size and shape with roughly similar openings to the Scentry ones? With similar amount of red visual stimuli?  Please rewrite this so that it is really clear what you actually did.  This needs to be clearly stated.

Response: This paragraph has been modified to During the two years study period, four experimental units were randomly arranged within a single block. A total of two blocks were assigned with replications made annually. The treatments included weed fabric mulch, shortleaf pine needle mulch, pine bark mulch, and untreated control. The four treatments randomly arranged within each block. Three mature blueberry buffer plants were used to separate each treatment (the average planting distance between the plants was 1.5 m). Adults SWD activity was monitored using two attractant lures used alongside Scentry® traps (Scentry Biological Inc., Billings, MT, USA). Trap 1 contained a Scentry bait pouch lure (herein referred to as bait trap) while traps 2 contain a liquid bait lure (herein referred to as Suzukii bait trap). The traps were distributed over an area of 137 m² at each sample site. Adult SWD caught in a trap in each treatment were recorded over a period of twelve weeks. The experiment took place in the Spring months (March – May) in 2017 and 2018.

“trap is mostly red externally with the lure hanging inside, small holes at the sides” I think that it is important to paint a better picture of the trap for the reader.  It is a clear jar covered with a primarily bright red label.  Add the city and state for the suppliers, please.

Response: Several sentences has been added in the text to clarify this.

Great Lakes IPM, should have city and state
Provide the company names, city and state for the two types of traps please

Response: The complete name with State and Country has been inserted.

“The experiments included two recommended traps” I don’t think that you mean that the experiments included the traps but that the numbers of SWD in each treatment were monitored by two different types of traps.

Response: We only used on type of trap and its complete name has been provided.

Suggested wording in title for table 1. “Average numbers of adult D. suzukii caught on blueberries with different mulching practices” rather than “Average numbers of adult D. suzukii caught on different mulching practices in blueberries”

Response: Modified throughout the text in Tables. 

You have an inconsistent use of the ® after Suzukii trap in the manuscript. 

Response: This has been modified to Suzukii bait trap for consistency.

You stated that by week 11 and 12 the blueberries may have been harvested.  When is the time of optimal attraction of SWD to blueberries?  Presumably it is when they are ripening so that earlier in the season when the berries are small, there may be fewer volatiles to attract and retain the flies in the area and that would reflect lower numbers as well.  I wonder how the statistical comparisons would look if you analyzed the data only making comparisons from weeks when blueberries were ripening or when your overall numbers (weekly collection) exceeded maybe 30 or so.  I think that the initial low collections are complicating your statistical comparisons. The high SE results from including early and late season data that have very low collections. I think that if examine some specific dates when flies are a high populations, that you would see difference in collections between the mulch types.

Response: The blueberry fruits mature over time during the fruiting season. Most of the fruits are harvested within three months period. Because fruits ripe at different timings, the fly attack is more noticeable while they visit the mature fruits to lay eggs.

Do the larvae not pupate in the mulch?  It may be that they do better pupating in the pine needle mulch than the others.

Response: This was the main idea to create a hostile condition for the full-grown larvae so that either they do not pupate or have difficulty in pupating. The larvae may not pupate in the weed fabric mulch; however, they may pupate in the plant based mulches.

Figure legends.  ‘Effects of three mulching practices on the capture of ….’ Should be ‘Effects of three mulching practices in comparison to unmulched controls on the capture of….’

Response: As advised, this has been modified in all Figures.

Reviewer 3 Report

Tables 3 and 4 do not have a clear explanation. And it should be written more comprehensibly.

Author Response

Comments and Suggestions for Authors

Tables 3 and 4 do not have a clear explanation. And it should be written more comprehensibly.
Response: Additional explanation for the Tables 3 and 4 is inserted.

Round 2

Reviewer 1 Report

The queries raised were well addressed. But I still feel that there are flaws in the research design. Drosophila is an actively flying insect and the buffer zone created was just 6 meters. There are reports stating that Drosophila can cover long distances. There is a high chance of error in sampling. This might be the reason for not getting statistical significance in most results. 

Emphasize this limitation in the manuscript (conclusion and discussion part)

Author Response

Reviewer 1: The queries raised were well addressed. But I still feel that there are flaws in the research design. Drosophila is an actively flying insect and the buffer zone created was just 6 meters. There are reports stating that Drosophila can cover long distances. There is a high chance of error in sampling. This might be the reason for not getting statistical significance in most results. Emphasize this limitation in the manuscript (conclusion and discussion part)

Response: Thank you for the comments and suggestions. As advised, we have included the following additional information in the Discussion and Conclusion sections:

Discussion: Also, short buffer zones (6 m) were kept in between the treatments under this study, and an earlier study could be a limiting factor to provide a clear picture of the actual adult flies caught as affected by each treatment [27]. In another study, a buffer zone of 15 meters was kept between the treatments to capture adult flies of D. suzukii [28].

Conclusion: However, this study provides preliminary information on the effects of three selected mulches on the capture of Drosophila suzukii (a highly mobile insect) due to the very short buffer zone between the treatments. Therefore, additional study in the bigger and open blueberry fields is necessary to elaborate on these results.
